# New Imatinib Derivatives with Antiproliferative Activity against A549 and K562 Cancer Cells

**DOI:** 10.3390/molecules27030750

**Published:** 2022-01-24

**Authors:** Andressa Oliveira, Stefany Moura, Luiz Pimentel, João Neto, Rafael Dantas, Floriano Silva-Jr, Monica Bastos, Nubia Boechat

**Affiliations:** 1Laboratório de Sintese de Farmacos, Instituto de Tecnologia em Farmacos-Farmanguinhos, Rua Sizenando Nabuco 100, Manguinhos, Rio de Janeiro 21041-250, RJ, Brazil; andressa.oliveira@far.fiocruz.br (A.O.); stefanybazan6@far.fiocruz.br (S.M.); luiz.pimentel@far.fiocruz.br (L.P.); monica.bastos@far.fiocruz.br (M.B.); 2Programa de Pós-graduação em Farmacologia e Química Medicinal do Instituto de Ciências Biomédicas, Centro de Ciências da Saúde, Bloco J, Ilha do Fundão, Rio de Janeiro 21941-902, RJ, Brazil; floriano@ioc.fiocruz.br; 3Laboratório de Bioquímica Experimental e Computacional de Fármacos, Instituto Oswaldo Cruz, FIOCRUZ, Av. Brasil 4365, Manguinhos, Rio de Janeiro 21040-360, RJ, Brazil; joao.neto@ioc.fiocruz.br (J.N.); rafael.dantas@ioc.fiocruz.br (R.D.)

**Keywords:** tyrosine kinase inhibitors, imatinib, K562, A549, PAPP and isatin

## Abstract

Tyrosine kinase enzymes are among the primary molecular targets for the treatment of some human neoplasms, such as those in lung cancer and chronic myeloid leukemia. Mutations in the enzyme domain can cause resistance and new inhibitors capable of circumventing these mutations are highly desired. The objective of this work was to synthesize and evaluate the antiproliferative ability of ten new analogs that contain isatins and the phenylamino-pyrimidine pyridine (PAPP) skeleton, the main pharmacophore group of imatinib. The 1,2,3-triazole core was used as a spacer in the derivatives through a click chemistry reaction and gave good yields. All the analogs were tested against A549 and K562 cells, lung cancer and chronic myeloid leukemia (CML) cell lines, respectively. In A549 cells, the 3,3-difluorinated compound (**3a**), the 5-chloro-3,3-difluorinated compound (**3c**) and the 5-bromo-3,3-difluorinated compound (**3d**) showed IC_50_ values of 7.2, 6.4, and 7.3 μM, respectively, and were all more potent than imatinib (IC_50_ of 65.4 μM). In K562 cells, the 3,3-difluoro-5-methylated compound (**3b**) decreased cell viability to 57.5% and, at 10 µM, showed an IC_50_ value of 35.8 μM (imatinib, IC_50_ = 0.08 μM). The results suggest that **3a**, **3c**, and **3d** can be used as prototypes for the development of more potent and selective derivatives against lung cancer.

## 1. Introduction

Protein tyrosine kinases (PTKs) are a group of approximately 90 enzymes that play essential roles in cells, such as in the regulation of cell division, cell differentiation and morphogenesis [1]. PTKs can be divided into receptor tyrosine kinases (RTKs) and nonreceptor tyrosine kinases (NRTKs). Examples of RTKs and NRTKs are insulin receptors and growth factor receptors (GFRs), such as epidermal growth factor receptor (EGFR) and ABL1, respectively [2]. The deregulated action of these PTKs is directly related to the development of some types of cancer [3].

PTK BCR-ABL1 is not expressed in healthy organisms because it is a product of cellular deregulation, and this PTK has been described as an oncogene that is present in 95% of patients with chronic myeloid leukemia (CML) [4]. The discovery of the relationship between this TK and CML made the development of imatinib (**1**) possible, as the first drug to be used against the PTK BCR-ABL1, which revolutionized the treatment of CML [5]. This drug acts by competitively inhibiting ATP binding to the BCR-ABL1 enzyme, preventing substrate phosphorylation, blocking its activity, and avoiding transduction of the signals essential for cellular functions. However, the emergence of cases of resistance to this drug has shown the need to develop new second- and third-generation inhibitors. However, even these new tyrosine kinase inhibitors (TKIs) have shown many cases of resistance, highlighting the need for a continuous search for new compounds that can treat these resistant tumors [6,7,8].

EGFR is a PTK receptor, also known as ErbB1/HER1, that belongs to the ErbB family, which also includes ErbB2/HER2/Neu, ErbB3/HER3 and ErbB4/HER42. EGFR participates in cell proliferation and apoptosis and has been classified as a proto-oncogene because it is commonly seen in cancers, such as non-small-cell lung cancer, metastatic colorectal cancer, glioblastoma, head and neck cancer, pancreatic cancer and breast cancer [9]. EGFR is a key mediator that plays a crucial role in cell growth, proliferation, survival and migration. This protein belongs to the kinase family and, in recent years, the discovery of the relationship between the overexpression of EGFR and solid tumors has made EGFR a target in modern medicinal chemistry for the planning of new anticancer agents [10].

The demand imposed by frequent mutations occurring in the kinase domain is still among the greatest limitations to treatment by target therapy, especially in oncology [11]. Therefore, it is imperative to develop new TKIs that can reduce both known resistances and others that may arise.

By continuing the work that the group has been developing to obtain new imatinib analogs [12], in this work, we synthesized a new series of ten imatinib analogs (**2a–e** and **3a–e**, Figure 1) and evaluated their cytotoxic activity in two human cancer cell lines: K562 (CML) and A549 (lung cancer). The compounds were designed as molecular hybrids containing phenylamino-pyrimidine pyridine (PAPP, in blue) and isatin (in red) scaffold units, connected by a 1*H*-1,2,3-triazole ring (in purple) (Figure 1). PAPP is an important pharmacophoric fragment of imatinib [13,14]. Isatins and some derivatives have relevance as TKIs [15], as shown for the cytotoxic activity of compound **4** against K562 cells, a cell lineage that has the constitutive activity of the TK BCR-ABL1. Compound **4** showed an IC_50_ of 0.03 μM, which was comparable to that obtained with the standard drug, irinotecan (CPT-11, IC_50_ = 0.07 μM) (Figure 1) [16]. Moreover, isatin scaffolds are present in sunitinib (**5**) and nintedanib (**6**), which are important TKIs. In derivatives **2a–e** and **3a–e**, the isatins have different substituents, which were chosen to verify their electronic contributions as electron attractor (Cl, Br, F) and electron donor (CH_3_) groups. The 1*H*-1,2,3-triazole ring was used as a spacer between the two units, PAPP and isatin, since this nucleus has been described in compounds with potential TK inhibitory activity, as in compound **7** [17].

The cell lines used herein (K562 and A549) are sensitive to imatinib. Although there is a vast literature showing the cytotoxicity of imatinib analogs in K562 cells, little is known about their effects in the A549 cell line. Shijie and coworkers showed that imatinib and its derivatives presented good inhibitory results in A549 cells. An example is compound **8**, which has a PAPP skeleton and decreased cell viability to 38.5% at a concentration of 150 μM (Figure 1) [18]. This result encouraged us to also test our compounds against this cell line.

## 2. Results and Discussion

### 2.1. Chemistry

Ten new hybrids presenting the PAPP group and isatin or derivatives spaced with a 1*H*-1,2,3-triazole ring were synthesized, as shown in Figure 1.

Initially, commercially available isatins (**11a–e**) were *N*-alkylated with propargyl bromide to generate 1-propyl-indoline-2,3-dione derivatives (**12a–e**) at an 83–97% yield (Figure 1). Compounds **12a–e** were characterized, and the data were in agreement with the literature [19,20]. *N*-(5-Azido-2-methylphenyl)-4-(pyridin-3-yl)pyrimidin-2-amine (**10**) was prepared via an aromatic nucleophilic substitution reaction, through the formation of a diazonium salt using intermediate **9**. This compound was obtained at a 98% yield and was characterized; the data were in agreement with the literature [19].

The fluorination of compounds **11a–e** with diethylaminosulfurtrifluoride (DAST) in CH_2_Cl_2_ at 25 °C produced intermediates **13a–e** after 4 h, with a 68–88% yield. The IR spectra of intermediates **13a–e** showed absorption bands in the region from 3200 to 3271 cm^−1^, corresponding to the axial strain of the NH bond, and stretches between 1268 and 1335 cm^−1^, corresponding to the axial strain of the CF_2_ bond [19]. Spectroscopic data of intermediates **13a–d** have already been described in the literature, but that of the intermediate **13e** is unpublished. The ^1^H NMR spectrum of intermediate **13e** showed a singlet at 11.20 ppm corresponding to a N-H hydrogen. In the ^13^C NMR spectrum, a triplet at 165.7 ppm with *J* = 29 Hz, corresponding to a carbonyl carbon and a triplet at 110.8 ppm with *J* = 247.9 Hz, corresponding to the C-3 carbon were observed, and this multiplicity is due to carbon-fluorine coupling with the difluoromethylene group. The ^19^F NMR spectrum showed a chemical shift at −111.1 ppm, corresponding to CF_2_, and the peak at 119.0 ppm corresponded to the C-F connection.

*Gem*-difluorinated intermediates **14a–e** were prepared at a 52–94% yield by the *N*-alkylation of compounds **13a–e**. The IR spectra of intermediates **14a–e** showed absorption bands in the region from 1276 to 1335 cm^−1^, corresponding to the axial deformation of CF_2_. For compounds **14d–e**, NMR analyses were performed, as they have not yet been described in the literature. In the ^1^H NMR analysis of intermediates **14d** and **14e**, it is possible to observe a signal at 7.30 or 7.37 ppm, related to H-7, and a doublet at 4.60 or 4.61 ppm, with a coupling constant of 2.5 Hz for H-1′. In the ^13^C NMR analyses of the intermediates **14d–e**, signals corresponding to C-1’, C-2’ and C-3’ were observed at 29.5 ppm, 76.5 ppm, and between 75.5 and 75.6 ppm, respectively. The ^19^F NMR spectrum showed chemical shifts between −110.6 and −110.7 ppm corresponding to CF_2_.

The 1,3-dipolar cycloaddition reaction between the azide derivative (**10**) and the terminal alkynes **12a–e** and **14a–e,** was performed via click chemistry conditions, using sodium ascorbate and copper sulfate in THF/H_2_O (1:1) at room temperature for 3 h, to obtain only one 1,4-regioisomer of the final products **2a–e** and **3a–e**, respectively.

New hybrids **2a–e** and **3a–e** were obtained in good yields, although some unsatisfactory yields may be associated with the low solubility of the final compounds, making their purification process difficult. The IR spectra for compounds **2a–e** showed absorption bands between 3383 cm^−1^ and 3440 cm^−1^, corresponding to the axial deformation of the N-H bond; 1736 and 1739 cm^−1^, corresponding to the axial deformation of the ketone carbonyl; and 1579 and 1618 cm^−1^, corresponding to the deformation of an amylic carbonyl. Compounds **3a–e** showed absorption bands between 1284 and 1298 cm^−1^, corresponding to the axial deformation of the CF_2_ bond. The ^1^H NMR analysis of compounds **2a****–e** and **3a–e** showed that methylene hydrogens were present and simple, with displacements between 2.33 and 2.43 ppm (see Appendix A). The CH_2_ hydrogens presented as simple with chemical shifts between 5.04 and 5.10 ppm. The hydrogens of the triazole ring characteristic of these compounds were observed as singlets in the region of 8.02 to 8.89 ppm (see Appendix A).

### 2.2. In Vitro Biological Evaluation

Biological evaluations of compounds **2a–e** and **3a–e** were performed in K562, A549 and WSS-1 cells. As previously described, K562 is a CML cell line, A549 is a human pulmonary carcinoma epithelial cell line, and WSS-1 is a healthy human cell line. WSS-1 cells were used as a reference for the calculation of the selectivity index (SI).

#### 2.2.1. Cytotoxic Effects in K562 and WSS-1 Cells

Compound screening in K562 cells revealed that compound **3b** had the highest cytotoxic activity among these imatinib analogs at 10 µM, presenting a percentage of viability of 57.5%, whereas imatinib showed 30.7% on under the same conditions. Subsequently, a concentration-response curve was carried out for compound **3b**, and an IC_50_ value of 35.8 μM was determined (imatinib, IC_50_ = 0.08 μM) (Table 1). In WSS-1 cells, compound **3b** showed an IC_50_ of 69.3 μM (imatinib, IC_50_ = 9.6 μM), resulting in an SI value of 1.9 (imatinib, SI = 120) (Figure 2 and Table 1).

Table 1 shows the IC_50_ and SI values of imatinib and its most cytotoxic analog, **3b,** on K562 cells.

#### 2.2.2. Cytotoxic Effects in A549 and WSS-1 Cells

The evaluations performed with A549 cells showed that compounds **3a**, **3c,** and **3d** reduced cell viability by 24.6%, 34.0%, and 49.3% at 10μM, respectively (Figure 3). Subsequently, the concentration-response curves were constructed, and derivatives **3a**, **3c,** and **3d** exhibited IC_50_ values of 7.2 μM, 6.4 μM, and 7.3 μM, respectively, proving to be approximately 10-fold more potent than imatinib (IC_50_ = 65.4 μM) (Table 2).

In WSS-1 cells, **3a**, **3c,** and **3d** showed IC_50_ values of 11.6, 13.5 and 18.6 μM (imatinib, IC_50_ = 9.6 μM) and SI values of 1.6, 2.1 and 2.5, respectively, and were up to 25-fold more selective than imatinib (SI = 0.1). Thus, in A549 cells, the new compounds **3a**, **3c** and **3d** were more potent and selective than imatinib (Figure 3 and Table 2). Imatinib was used as a standard, even though it is not a drug used to treat lung cancer. This was due to the good results obtained by Shijie and coworkers, which decreased A549 cells viability to 38.8% at 150 μM (Figure 1) [18]. In addition, the literature states that imatinib can be used as a potential treatment for NSCLC, as it was able to inhibit the growth of A549 cells with an IC_50_ value in the range of 2–3 μM [20].

Table 2 shows the IC_50_ and SI values of imatinib and its most cytotoxic analogs (**3a**, **3c** and **3d**) in A549 cells.

The study of the relationship between the structure and the biological activity of the synthesized compounds showed the importance of the CF_2_ group since, in compounds **2a–e**, the absence of this group showed a loss of activity. In addition, according to the biological tests performed, the intrinsic characteristics of the substituents on the C-5 carbon of the isatin-derived ring do not seem to influence the level of activity.

### 2.3. Kinase Inhibition Assay

The compounds **2a–e** and **3a–e** did not show ABL1-inhibitory activity at 0.5 or 10 μM under the given assay conditions. A possible explanation for this result is that the compounds may have a different mechanism of action from imatinib, which showed a subsequent percentage inhibition at 0.5 and 10 µM in the same assay. Another hypothesis could be a probable higher affinity of the substrate to the enzyme than the analogs, which may have interfered with the interaction of the latter with this kinase. Thus, more experiments are needed to characterize the mechanism of these compounds, by varying the enzyme inhibition assay conditions.

## 3. Materials and Methods

### 3.1. Chemistry

All reagents and solvents used were of analytical grade. Briefly, ^1^H, ^13^C and ^19^F nuclear magnetic resonance (NMR) spectra were generated at 400.00, 100.00 and 376.00 MHz, respectively, at 25 °C using a Bruker Avance III HD instrument (Bruker AG, Fällanden, Switzerland) equipped with a prodigy BBO 400 S1 probe (Bruker AG, Fällanden, Switzerland). Tetramethylsilane was used as an internal standard. The chemical shifts (δ) are reported in ppm, and the coupling constants (*J*) are reported in Hertz. Fourier transform infrared (FTIR) absorption spectra were recorded on a Thermo Scientific spectrophotometer (Nicolet 6700, Thermo Fisher Scientific, Waltham, MA, USA). The melting point (m.p.) values were determined using a Büchi model B-545 apparatus (Büchi Corporation, Flawil, Switzerland). Thin-layer chromatography (TLC) was performed using Merck TLC silica gel 60 F254 aluminum sheets (Merck KGaA, Darmstadt, Germany), 20 × 20 cm (eluent hexane/ethyl acetate 2:8). A mass spectrometry system was coupled to obtain electron impact gas chromatography (MS-GC) spectra at 70 eV on an Agilent 6890 apparatus, with an Agilent 5973 mass spectrometer (Agilent, Santa Clara, CA, USA). Low-resolution mass spectra were obtained by electrospray ionization (MS-ESI) on a Micromass ZQ4000 apparatus (Waters, Milford, MA, USA). High-resolution mass spectra (HR-MS) were registered using electron ionization mass spectrometry (EI-MS, digitalizing ES + capillary) on a QTOF Compact (Brucker AG, Fällanden, Switzerland).

#### 3.1.1. General Procedure for the Preparation of 1-((1-(4-methyl-3-((4-pyridin-3-yl)pyrimidin-2-yl)amine)phenyl)-1*H*-1,2,3-triazol-4-yl) methyl) indolin-2,3-dione (**2a–e**)

Briefly, 2.3 mmol (1.3 eq) of the corresponding acetylenes, **12a–e**, 1.8 mmol (1 eq) of the *N*-(5-azido-2-methylphenyl)-4-(pyridin-3-yl)pyrimidin-2-amine (**10**), 0.1 mmol (1 eq) of sodium ascorbate 0.1 mmol (0.06 eq) of copper sulfate, and 20 mL of THF and water (1:1) were added to a monotubulated flask. The reaction was kept under magnetic stirring at room temperature for 3 h. The progress of the reaction was monitored using TLC (hexane/ethylacetate 3:7). On completion, 30 mL of water was added and the mixture was extracted with CHCl_3_ (3 × 50 mL). The organic phase was dried with anhydrous Na_2_SO_4,_ and the solvent was removed by evaporation. The residual crude product was purified via silica gel column chromatography, using the gradient mixture of chloroform: methanol (9.5:0.5).

The attributes can be summarized as follows:

**1-((1-(4-methyl-3-((4-(pyridin-3-yl)pyrimidin-2-yl)amino)phenyl)-1*H*-1,2,3-triazol-4-yl)methyl)indoline-2,3-dione (2a):** Yield: 58%; m.p.: 237–239 °C. IR (cm^−1^): 1739; 1618; 1578; 755; 709; 656. MS-ESI [M+Na]^+^ (%): 511 (100). ^1^H-NMR (400 MHz, CD_3_COCD_3_, δ, ppm): 2.33 (s, 3H, H-26); 5.07 (s, 2H, H-8); 7.13 (td, *J* = 0.7Hz, 1H, H-6); 7.20 (d, *J* = 7.9 Hz, 1H, H-7); 7.42 (d, *J* = 8.5 Hz, 1H, H-14); 7.51 (m, 3H, H-13, H-19, H24); 7.58 (dd, *J* = 0.8 Hz, 1H, H-5); 7.62 (td, *J* = 1.3Hz, 1H, H-4); 8.28 (d, *J* = 2.2Hz, 1H, H-12); 8.47 (dt, *Jo =* 8.4 Hz, *Jm =* 2.0 Hz, 1H, H-25); 8.56 (d, *J* = 5.2 Hz, 1H, H-18); 8.69 (dd, *J* = 1.6 Hz, 1H, H-23); 8.82 (s, 1H, H-10); 9.13 (s, 1H, NH); 9.27 (d, *J* = 1.6 Hz, 1H, H-22). ^13^C-NMR (100 MHz, CD_3_COCD_3_, δ, ppm):17.7; 34.9; 108.2; 110.9; 114.9; 115.4; 117.6; 121.5; 123.3; 123.7; 124.3; 131.2; 131.9; 134.3; 134.4; 137.9; 138.9; 142.6; 147.9; 149.9; 151.3; 157.7; 159.5; 160.6; 161.5; 182.9. HR-MS (IES)^+^: Calcd for C_27_H_20_N_8_O_2_: 488.1709, found: 489.1782. Elemental analysis CHN: Calcd for C_27_H_20_N_8_O_2_ (%): C, 66.38; H, 4.13; N, 22.94, found (%): C, 66.56; H, 4.45; N, 22.82.

**5-methyl-1-((1-(4-methyl-3-((4-(pyridin-3-yl)pyrimidin-2-yl)amino)phenyl)-1*H*-1,2,3-triazol-4 yl)methyl)indoline-2,3-dione(2b):** Yield: 41%; m.p.: 235-237 °C. IR (cm^−1^): 1736; 1580; 1532; 789; 658. MS-ESI [M-H]^+^ (%): 501 (100). ^1^H-NMR (400 MHz, CD_3_COCD_3_, δ, ppm): 2.26 (s, 3H, CH_3_-5); 2.33 (s, 3H, H-26); 5.04 (s, 2H, H-8); 7.08 (d, *J* = 8.0 Hz, 1H, H-7); 7.41 (m, 3H, H-13, H-14, H-19); 7.53 (m, 3H, H-4, H-6, H-24); 8.28 (d, *J* = 2.2 Hz, 1H, H-12); 8.50 (dt, *Jo =* 8.1 Hz, *Jm =* 1.8 Hz, 1H, H-25); 8.56 (d, *J* = 5.2 Hz, 1H, H-18); 8.71 (d, *J* = 2.4 Hz, 1H, H-23); 8.81 (s, 1H, H-10); 9.15 (s, 1H, NH); 9.28 (s, 1H, H-22). ^13^C-NMR (100 MHz, CD_3_COCD_3_, δ, ppm): 17.7; 19.9; 34.9; 108.2; 110.8; 114.9; 115.4; 117.5; 121.4; 123.9; 124.6; 131.2; 131.9; 132.6; 134.4; 134.7; 138.3; 138.9; 142.6; 147.6; 147.8; 150.9; 157.8; 159.5; 160.6; 161.3; 183.2. HR-MS (IES)^+^: Calcd for C_28_H_22_N_8_O_2_: 502.1866, found: 503.1938. Elemental analysis CHN: Calcd for C_28_H_22_N_8_O_2_: C. 66.92; H. 4.41; N. 22.30, found (%): C. 66.31; H. 4.50; N. 22.64.

**5-chloro-1-((1- (4-methyl-3-((4-(pyridin-3-yl)pyrimidin-2-yl)amino)phenyl)-1*H*-1,2,3-triazol-4 yl)methyl)indoline-2,3-dione(2c):** Yield: 50%; m.p.: 257-260 °C. IR (cm^−1^): 1739; 1579; 1531; 809; 656. MS-ESI [M-H]^−^ (%): 521 (100), ^1^H-NMR (400 MHz, CD_3_COCD_3_, δ, ppm): 2.33 (s, 3H, H-26); 5.07 (s, 2H, H-8); 7.23 (d, *J* = 8.4 Hz, 1H, H-7); 7.42 (d, *J* = 8.4 Hz, 1H, H-6); 7.51 (m, 2H, H-13, H-19); 7.55 (dd, *J* = 4.8 Hz, 1H, H-24); 7.66 (m, 2H, H-4, H-14); 8.27 (d, *J* = 2.2 Hz, 1H, H-12); 8.50 (dt, *Jo =* 8.1 Hz, *Jm =* 1.6 Hz,. 1H, H-25); 8.56 (d, *J* = 5.2 Hz, 1H, H-18); 8.71 (s, 1H, H-23); 8.75 (s, 1H, H-10); 9.15 (s, 1H, NH); 9.28 (s, 1H, H-22). ^13^C-NMR (100 MHz, CD_3_COCD_3_, δ, ppm): 17.7; 35.1; 108.2; 112.7; 114.9; 115.4; 119.0; 121.5; 123.8; 123.9; 127.5; 131.3; 131.9; 134.4; 134.8; 136.8; 138.9; 142.4; 147.5; 148.4; 150.9; 157.5; 159.5; 160.6; 161.3; 181.8. HR-MS (IES)^+^: Calcd for C_27_H_19_ClN_8_O_2_: 522.1319, found: 523.1392. Elemental analysis CHN: Calcd for C_27_H_19_ClN_8_O_2_: C. 62.01; H. 3.66; N. 21.43, found (%): C. 62.31; H. 3.14; N. 21.65.

**5-bromo-1-((1-(4-methyl-3-((4-(pyridin-3-yl)pyrimidin-2-yl)amino)phenyl)-1*H*-1,2,3-triazol-4 yl)methyl)indoline-2,3-dione (2d):** Yield: 50%; m.p.: 267-270 °C. IR (cm^−1^): 1736; 1579; 1531; 808; 653. MS-ESI [M+Na]^+^ (%): 589 (80). ^1^H-NMR (400 MHz, CD_3_COCD_3_, δ, ppm): 2.33 (s, 3H, H-26); 5.07 (s, 2H, H-8); 7.18 (d, *J* = 8.4 Hz, 1H, H-7); 7.42 (d, *J* = 8.4 Hz, 1H, H-6); 7.51 (m, 2H, H-13, H-19); 7.56 (dd, *J* = 5.0 Hz, 1H, H-24); 7.74 (d, *J=* 2.1 Hz, 1H, H-14); 7.80 (dd, *J* = 2.1 Hz, 1H, H-4); 8.27 (d, *J* = 2.2 Hz, 1H, H-12); 8.51 (dt, *Jo =* 8.0 Hz, *Jm =* 1.8 Hz, 1H, H-25); 8.56 (d, *J* = 5.2 Hz, 1H, H-18); 8.71 (dd, *J* = 1.4 Hz, 1H, H-23); 8.80 (s, 1H, H-10); 9.15 (s, 1H, NH); 9.2 (d, *J* = 1.56 Hz, 1H, H-22). ^13^C-NMR (100 MHz, CD_3_COCD_3_, δ, ppm): 17.7; 35.0; 108.2; 113.1; 115.0; 115.4; 119.0; 121.5; 123.9; 126.6; 131.3; 131.9; 132.1; 134.4; 134.9; 138.9; 139.6; 142.4; 147.5; 148.8; 150.8; 157.3; 159.5; 160.6; 161.3; 181.7. HR-MS (IES)^+^: Calcd for C_27_H_19_BrN_8_O_2_: 566.0814, found: 567.0887. Elemental analysis CHN: Calcd for C_27_H_19_BrN_8_O_2_: C. 57.15; H. 3.38; N. 19.75, found (%): C. 57.00; H. 3.21; N. 19.77.

**5-fluoro-1-((1-(4-methyl-3-((4-(pyridin-3-yl)pyrimidin-2-yl)amino) phenyl)-1*H*-1,2,3-triazol-4 yl) methyl)indoline-2,3-dione (2e):** Yield: 82%; m.p.: 251-253 °C. IR (cm^−1^): 1736; 1581; 1537; 809. MS-ESI [M-H]^+^ (%): 505 (70). ^1^H-NMR (400 MHz, CD_3_COCD_3_, δ, ppm): 2.33 (s, 3H, H-26); 5.07 (s, 2H, H-8); 7.22 (dd, *J* = 3.7 Hz, 1H, H-7); 7.43 (d, *J* = 8.4 Hz, 1H, H-14); 7.50 (m, 5H, H-4, H-6, H-13, H-19, H-24); 8.27 (d, *J* = 2.0 Hz, 1H, H-12); 8.50 (dt, *Jo =* 8.2 Hz, *Jm =* 1.9 Hz, 1H, H-25); 8.55 (d, *J* = 5.2 Hz, 1H, H-18); 8.68 (dd, *J* = 1.5 Hz, H-23); 8.81 (s, 1H, H-10); 9.14 (s, 1H, NH); 9.26 (d, *J* = 1.7 Hz, 1H, H-22). ^13^C-NMR (100 MHz, CD_3_COCD_3_, δ, ppm): 17.8; 35.1; 108.3; 111.5 (d, *J* = 24.4Hz); 112.5 (d, *J* = 7.5Hz); 115.1; 115.5; 118.7 (d, *J* = 7.14 Hz); 121.6; 123.8; 124.1; 131.4; 132.0; 134.4; 134.5; 139.0; 142.6; 146.3; 148.1; 151.5; 158.5 (d, *J* = 239.8 Hz); 157.9 (d, *J* = 1.0 Hz); 159.6; 160.7; 161.6; 182.4 (d, *J* = 1.3 Hz). ^19^F-NMR (376 MHz, DMSO, δ, ppm): −119.9. HR-MS (IES)^+^: Calcd for C_27_H_19_FN_8_O_2_: 506.1615, found: 507.1687. Elemental analysis Calcd for C_27_H_19_FN_8_O_2_: C. 64.03; H. 3.78; N. 22.12, found (%): C. 64.68; H. 3.41; N. 22.62.

#### 3.1.2. General Procedure for the Preparation of 3.3-difluoro-1-((1-(4-methyl-3-((4-(pyridine-3-yl)pyrimidin-2-yl)amino)phenyl)-1H-1,2,3-triazole-4-yl)methyl)indoline-2-one (**3a–e**)

Briefly, 2.3 mmol (1.3 eq) of the corresponding acetylenes **14a–e**, 1.8 mmol (1 eq) of the *N*-(5-azido-2-methylphenyl)-4-(pyridine-3-yl)pyrimidin-2-amine (**10**), 0.1 mmol (1 eq) of sodium ascorbate 0.1 mmol (0.06 eq) of copper sulfate and 20 mL of THF and water (1:1) were added in a monotube flask. The reaction was kept under magnetic stirring at room temperature for 3 h. The progress of the reaction was monitored using TLC (hexane/ethyl acetate 3:7). On completion, 30 mL of water was added, and the mixture was extracted with CHCl_3_ (3 × 50 mL). The organic phase was dried with anhydrous Na_2_SO_4_, and the solvent was removed by evaporation. The residual crude product was purified via silica gel column chromatography using the gradient mixture chloroform: methanol (9.5:0.5).

The analyses can be summarized as follows:

**3.3-difluoro-1-((1-(4-methyl-3-((4-(pyridin-3-yl)pyrimidin-2-yl)amino)phenyl)-1*H*-1,2,3-triazole-4-yl)methyl)indoline-2-one (3a)**: Yield: 51%; m.p.: 222 °C. IR (cm^−1^): 3435; 3123; 1746; 1580; 1286. MS-ESI [M+1]^+^ (%): 511 (100).^1^H-NMR (400 MHz, CD_3_COCD_3_, δ, ppm): 2.34 (s, 3H, H-26); 5.09 (s, 2H, H-8); 7.24 (t, *J* = 7.6 Hz, 1H, H-6); 7.34 (d, *J* = 7.9 Hz, 1H, H-7); 7.44 (d, *J* = 8.4 Hz, 1H, H-14); 7.50 (m, 2H, H-19, H-24); 7.58 (m, 2H, H-5, H-13); 7.72 (d, *J* = 7.4 Hz, 1H, H-4); 8.31 (d, *J* = 2.2 Hz, 1H, H-12); 8.47 (dt, *Jo =* 8.1 Hz, *Jm =* 1.6 Hz, 1H, H-25); 8.57 (d, *J* = 5.2 Hz, 1H, H-18); 8.70 (s, H, H-23); 8.89 (s, 1H, H-10); 9.15 (s, 1H, NH); 9.29 (s, 1H, H-22). ^13^C-NMR (100 MHz, CD_3_COCD_3_, δ, ppm): 17.8; 35.2; 108.3; 111.0 (t, *J* = 246.6 Hz); 111.4; 115.3; 115.7; 118.7 (t, *J* = 22.7 Hz); 121.9; 123.8; 124.0; 124.6; 131.3; 132.0; 134.2; 134.4; 139.0; 141.8; 142.9 (t, *J* = 7.0 Hz); 148.0; 151.4; 159.6; 160.7; 161.6; 164.0 (t, *J* = 29.7 Hz). ^19^F-NMR (376 MHz, DMSO, δ, ppm): −110.4. HR-MS (IES)^+^: Calcd for C_27_H_20_F_2_N_8_O: 510.1728, found: 511.1800. Elemental analysis CHN: Calcd for C_27_H_20_F_2_N_8_O: C. 63.52; H. 3.95; N. 21.95, found (%): C. 63.35; H. 3.81; N. 21.68.

**3,3-difluoro-5-methyl-1-((1-(4-methyl-3-((4-(pyridin-3-yl)pyrimidin-2-yl)amino)phenyl)-1*H*-1,2,3-triazol-4-yl)methyl)indolin-2-one (3b)**: Yield: 25%; m.p.: 102–104 °C. IR (cm^−1^): 1746; 1578; 1298. MS-ESI [M+Na]^+^ (%): 547 (100).^1^H-NMR (400 MHz, CD_3_COCD_3_, δ, ppm): 2.34 (s, 3H, H-26); 2.42 (s, 3H, H-5′); 5.07 (s, 2H, H-8); 7.16 (s, 1H, NH); 7.26 (m, 4H, H-6, H-7, H-14); 7.29 (m, 2H, H-19, H-24); 7.35 (s, 1H, H-13); 7.48 (dd, *J* = 4.8 Hz, 1H, H-4); 8.02 (s, 1H, H-10); 8.47 (dt, *Jo =* 8.4 Hz, *Jm =* 1.8 Hz, 1H, H-25); 8.55 (d, *J* = 5.2 Hz, 1H, H-18); 8.74 (d, *J* = 3.6 Hz, 1H, H-23); 8.94 (d, *J* = 2.0 Hz, 1H, H-12); 9.26 (d, *J* = 1.0 Hz, 1H, H-22). ^13^C-NMR (100 MHz, CD_3_COCD_3_, δ, ppm): 17.8; 20.9; 35.6; 109.0; 111.1 (t, *J* = 248.4 Hz); 110.9; 112.3; 114.2; 119.9 (t, *J* = 22.6 Hz); 120.9; 123.9; 125.2; 127.4; 131.3; 132.4; 133.9; 134.1; 138.7; 140.1 (t, *J* = 7.1 Hz); 142.2; 148.4; 151.8; 159.2; 160.1; 162.7; 165.2 (t, *J* = 30.6 Hz). ^19^F-NMR (376 MHz, DMSO, δ, ppm): −111.6. HR-MS (IES)^+^: Calcd for C_28_H_22_F_2_N_8_O: 524.1885, found: 525.1957. Elemental analysis CHN: Calcd for C_28_H_22_F_2_N_8_O: C. 64.12; H. 4.23; N. 21.36, found (%): C. 64.27; H. 3.99; N. 21.01.

**5-chloro-3,3-difluoro-1-((1-(4-methyl-3-((4-(pyridin-3-yl)pyrimidin-2-yl)amino)phenyl)-1*H*-1,2,3-triazol-4-yl)methyl)indoline-2-one****(3c)**: Yield: 45%; m.p.: 223–224 °C. IR (cm^−1^): 3435; 3123; 1754; 1580; 1286. MS-ESI [M+Na]^+^ (%): 567 (100). ^1^H-NMR (400 MHz, CD_3_COCD_3_, δ, ppm): 2.34 (s, 3H, H-26); 5.10 (s, 2H, H-8); 7.37 (d, *J* = 8.5 Hz, 1H, H-7); 7.44 (d, *J* = 8.4 Hz, 1H, H-14); 7.55 (m, 3H, H-13, H-19, H-24); 7.66 (d, *J* = 8.4 Hz, 1H, H-6); 7.94 (d, *J* = 1.8 Hz, 1H, H-4); 8.30 (d, *J* = 2.1 Hz, 1H, H-12); 8.47 (d, *J* = 7.9 Hz, 1H, H-25); 8.57 (d, *J=* 5.2 Hz, 1H, H-18); 8.72 (s, 1H, H-23); 8.87 (s, 1H, H-10); 9.15 (s, 1H, NH); 9.31 (s, 1H, H-22). ^13^C-NMR (100 MHz, CD_3_COCD_3_, δ, ppm): 17.7; 35.3; 108.2; 110.3 (t, *J* = 248.1 Hz); 113.0; 115.2; 115.6; 120.3 (t, *J* = 22.7 Hz); 121.8; 124.9; 128.1; 131.2; 131.9; 133.9; 134.3; 134.4; 138.9; 141.5; 141.7 (t, *J* = 6.7 Hz); 147.9; 151.2; 159.5; 160.6; 161.5; 163.7 (t, *J* = 29.6 Hz). ^19^F-NMR (376 MHz, DMSO, δ, ppm): −111.7. HR-MS (IES)^+^: Calcd for C_27_H_19_ClF_2_N_8_O: 544.1338, found: 545.1411. Elemental analysis CHN: Calcd for C_27_H_19_ClF_2_N_8_O: C, 59.51; H, 3.51; N, 20.56, found (%): C, 59.35; H, 3.35; N, 20.23.

**5-bromo-3.3-difluoro-1-((1-(4-methyl-3-((4-(pyridin-3-yl)pyrimidin-2-yl)amino)phenyl)-1*H*-1,2,3-triazol-4-yl)methyl)indoline-2-one****(3d)**:Yield: 78%; m.p.: 218–219 °C. IR (cm^−1^): 3056; 1753; 1579; 1284. MS-ESI [M+1]^+^ (%): 590 (100). ^1^H-NMR (400 MHz, CD_3_COCD_3_, δ, ppm): 2.34 (s, 3H, H-26); 5.09 (s, 2H, H-8); 7.30 (d, *J* = 8.5 Hz, 1H, H-7); 7.44 (d, *J* = 8.4 Hz, 1H, H-14); 7.51 (m, 2H, H-19, H-24); 7.55 (dd, *J* = 2.3 Hz, 1H, H-13); 7.80 (d, *J* = 8.5 Hz, 1H, H-6); 8.04 (d, *J* = 1.7 Hz,1H, H-4); 8.30 (d, *J* = 2.2 Hz, 1H, H-12); 8.46 (d, *J* = 8.0 Hz, 1H, H-25); 8.57 (d, *J* = 5.2 Hz, 1H, H-18); 8.70 (s, 1H, H-23); 8.87 (s,1H, H-10); 9.15 (s, 1H, NH); 9.29 (s, 1H, H-22). ^13^C-NMR (100 MHz, CD_3_COCD_3_, δ, ppm): 17.7; 35.2; 108.2; 110.2 (t, *J* = 248.1 Hz); 113.4; 115.2; 115.5; 115.7; 120.6 (t, *J* = 22.8 Hz); 121.8; 123.7; 127.5; 131.2; 131.9; 134.4; 136.8; 138.9; 141.5; 142.2 (t, *J* = 6.8 Hz); 148.0; 151.4; 159.5; 160.6; 161.5; 163.5 (t, *J* = 29.8 Hz). ^19^F-NMR (376 MHz, DMSO, δ, ppm): −110.7. HR-MS (IES)^+^: Calcd for C_27_H_19_BrF_2_N_8_O: 588.0833, found: 589.0906. Elemental analysis CHN: Calcd for C_27_H_19_BrF_2_N_8_O: C, 55.02; H, 3.25; N, 19.01, found (%): C, 54.97; H, 3.19; N, 19.60.

**(4-methyl-3-((4-(pyridin-3-yl)pyrimidin-2-yl)amino)phenyl)-1*H*-1,2,3-triazol-4-yl)methyl)indoline-2-one (3e)**: Yield: 30%; m.p.: 220 °C, IR (cm^−1^): 3737; 3055; 1748; 1578; 1291. MS-ESI [M+1]^+^ (%): 529 (100). ^1^H-NMR (400 MHz, CD_3_COCD_3_, δ, ppm): 2.43 (s, 3H, H-26); 5.07 (s, 2H, H-8); 7.15 (s, 1H, NH); 7.21 (m, 1H, H-7); 7.27 (m, 2H, H-14, H-24); 7.31 (m, 2H, H-13, H-19); 7.40 (dd, *J* = 3.8 Hz, 1H, H-6); 7.49 (dd, *J* = 4.8 Hz,1H, H-4); 8.05 (s, 1H, H-10); 8.47 (dt, *Jo* = 8.1 Hz, *Jm* = 1.9 Hz, 1H, H-25); 8.56 (d, *J* = 5.2 Hz, 1H, H-18); 8.76 (d, *J* =3.9 Hz, 1H, H-23); 8.95 (d, *J* = 1.9 Hz,1H, NH); 9.27 (s,1H, H-22). ^13^C-NMR (100 MHz, CD_3_COCD_3_, δ, ppm): 17.8; 35.7; 109.1; 112.2; 112.5; 112.6 (t, *J* = 25.6 Hz); 114.2; 120.3 (d, *J* = 23.4 Hz); 121.1; 127.5; 131.3; 132.4; 134.7; 135.3 (t, *J* = 297.0 Hz); 138.8; 141.8; 148.5; 151.8; 158.5 (d, *J* = 239.8 Hz); 159.2; 160.1; 162.8; 164.0 (t, *J* = 29.7 Hz). ^19^F-NMR (376 MHz, DMSO, δ, ppm): −111,9, −117,1. HR-MS (IES)^+^: Calcd for C_27_H_19_F_3_N_8_O: 528.1634, found: 529.1706. Elemental analysis CHN: Calcd for C_27_H_19_F_3_N_8_O: C, 61.36; H, 3.62; N, 21.20, found (%): C, 61.19; H, 3.39; N, 21.15.

#### 3.1.3. General Procedure for the Preparation of N-(5-azido-2-methylphenyl)-4-(pyridin-3-yl)pyrimidin-2-amine (**10**)

Briefly, 2-phenylaminopyrimidine (PAPP) (**9**) (1 eq) was added in 30 mL of water in a batch-flask under an ice bath; sulfuric acid was then added until the suspension became a solution. At this point, the sodium nitrites (1.5 eq) in 2 mL of water were added to the mixture, which was kept under magnetic stirring for 30 min, at −5 to 0 °C. Subsequently, sodium azide (2 eq) in 2 mL of water was added, and the mixture was kept under magnetic stirring for 10 min at −5 to 0 °C. Thereafter, the ice bath was removed, and the medium was left under magnetic stirring at room temperature for 3 h. The progress of the reaction was monitored using TLC (chloroform/methanol 9:1). On completion, the pH was adjusted to 7.0 with potassium carbonate solution and then extracted with dichloromethane (3 × 50 mL). The organic phase was dried with anhydrous Na_2_SO_4_, and the solvent was removed by evaporation. The product was obtained as a slightly yellowish solid.

Yield: 98%; m.p.: 115.3 °C. IR (cm^−1^): 3325; 2109; 2079; 1526; 790; 695; 687. MS-ESI [M-H]^+^ (%): 302 (100).

#### 3.1.4. General Procedure for the Preparation of 1-propyl-indoline-2,3-dione (**12a–e**)

Briefly, 6.8 mmol (1 eq) of the corresponding isatins (**11a–e**), 11 mmol (1.63 eq) of propargyl bromide, 12.92 mmol (1.9 eq) of potassium carbonate, 2.1 mmol (0.32 eq) of sodium iodide and 6 mL of anhydrous DMF were added to a monotube flask. The progress of the reaction was monitored using TLC (hexane/ethylacetate 7:3). After 2 h at room temperature, 5 mL of 3N HCl solution and ice were added, and a precipitate was formed. This solid was filtered by vacuum to give the desired product.

The attributes can be summarized as follows:

**1-propyl-indoline-2,3-dione (12a)**: Yield: 97%. m.p.: 144–146 °C (lit. 144–146 °C) [19]. IR (cm^−1^): 3242; 2124; 1725; 1617. MS-GC (70 eV, *m*/*z*, %): 129 (100); 185 (81); 102 (54); 90 (48); 128 (39). 

**5-methyl-1-propylindoline-2,3-dione (12b)**: Yield: 83%. m.p.: 147–150 °C (lit. 149–151 °C) [19]. IR (cm^−1^): 3243; 2126; 1725; 1616. MS-GC (70 eV, *m*/*z*, %): 199 (100), 143 (99); 142 (80); 115 (52); 116 (27).

**5-chloro-1-propylindoline-2,3-dione (12c)**: Yield: 90%. m.p.: 157–159 °C (lit. 158–159 °C) [19]. IR (cm^−1^): 3225; 2120; 1726; 1601. MS-GC (70 eV, *m*/*z*, %): 163 (100); 219 (92); 228 (68); 124 (46); 74 (51); 165 (34). 

**5-bromo-1-propylindoline-2,3-dione(12d)**: Yield: 90%. m.p.: 161–162 °C (lit. 157–158 °C) [21]. IR (cm^−1^): 3243; 2106; 1729; 1603. MS-GC (70 eV, *m*/*z*, %): 128 (100); 208 (84); 206 (83), 262 (82); 264 (81).

5**-fluoro-1-propylindoline-2,3-dione**
**(12e)**: Yield: 96%. m.p.: 148–152 °C (lit. 158–159 °C) [21]. IR (cm^−1^): 3225; 2121; 1726; 1762. MS-GC (70 eV, *m*/*z*, %): 203 (99); 147 (100); 120 (62); 108 (57); 146 (41).

#### 3.1.5. General Procedure for the Preparation of 3,3-difluoroindolin-2-one (**13a–e**)

Briefly, 3.4 mmol (1 eq) of the corresponding isatins **11a–e**, 17 mmol (5 eq) of DAST and 30 mL of CH_2_Cl_2_ were added into a monotube flask. The mixture was maintained at room temperature with magnetic stirring under a nitrogen atmosphere for 4 h. The progress of the reaction was monitored using TLC (hexane/ethyl acetate 7:3). The mixture was washed with water (3 × 40 mL), then the organic phase was dried with anhydrous Na_2_SO_4_, and the solvent was removed by evaporation. The product was obtained as a white solid.

The analyses can be summarized as follows:

**3,3-difluoroindolin-2-one (13a)**: Yield: 77%; 136–138 °C (lit. 137–139 °C) [21]. IR (cm^−1^): 3250; 1744; 1304. MS-ESI [M-1]^+^,(%): 168 (100). MS-GC (70 eV, *m*/*z*, %): 169 (100); 141 (92); 114 (86); 75 (10); 126 (10). 

**3,3-difluoro-5-methylindoline-2-one (13b)**: Yield: 88%; m.p.: 157–159 °C (lit. 155–157 °C) [22]. IR (cm^−1^): 3247; 1747; 1331. MS-GC (70 eV, *m*/*z*, %): 182 (100); 221 (94); 192 (17); 178 (13); 122 (13).

**5-chloro-3,3-difluoroindolin-2-one (13c)**: Yield: 68%; m.p.: 180–182 °C (lit. 183–185 °C) [21]. IR (cm^−1^): 3243; 1726; 1335. MS-GC (70 eV, *m*/*z*, %): 175 (100); 203 (83); 148 (60); 177 (32); 205 (27).

**5-bromo-3,3-difluoroindolin-2-one (13d)**: Yield: 75%; m.p.: 192–194 °C. IR (cm^−1^): 3200; 1726; 1268. HR-MS (IES)^−^: Calcd. for C_8_H_4_F_2_NO: 246.9400, found: 245.9371. NMR ^1^H (400 MHz; CD_3_COCD_3_, δ, ppm): 6.95 (dd, *J* = 1.5 Hz, 1H, H-7); 7.71 (m, 1H, H-6); 7.96 (m, 1H, H-4); 11.32 (s, 1H, NH), NMR ^13^C (100 MHz; CD_3_COCD_3_, δ, ppm): 110.5 (t, *J* = 248.6 Hz, C-3); 113.8 (C-7); 114.7 (t, *J* = 2.1 Hz, C-5); 121.0 (t, *J* = 22.8 Hz, C-4); 127.8 (C-6); 136.8 (C-7a); 141.8 (t, *J* = 7.4 Hz, C-3a); 165.2 (t, *J* = 29.1 Hz, C-2), NMR ^19^F (376 MHz; DMSO, δ, ppm): −111.0.

**5-fluoro-3,3-difluoroindolin-2-one (13e)**: Yield: 80%; m.p.: 184–186 °C. IR (cm^−1^): 3271; 1739; 1282. HR-MS (IES)^-^: Calcd. For C_8_H_4_F_3_NO: 187.0200, found: 186.0172. NMR^1^H (400 MHz; CD_3_COCD_3_, δ, ppm): 7.0 (m, 1H, H-7); 7.40 (m, 1H, H-6);7.70 (m, 1H, H-4); 11.20 (s, 1H, NH), NMR^13^C (100 MHz; CD_3_COCD_3_, δ, ppm): 110.8 (t, *J* = 247.9Hz, C-3); 112.7 (C-7); 113.1 (C-7a); 120.2 (td, *J* = 8.5Hz, C-4); 120.8 (d, *J* = 23.4 Hz, C-6); 138.7 (td, *J* = 2.1 Hz, C-3a); 158.2 (d, *J* = 238.8 Hz, C-5); 165.7 (t, *J* = 29.0 Hz, C-2), NMR^19^F (376 MHz; DMSO, δ, ppm): −111.1 (CF_2_); −119.0 (CF).

#### 3.1.6. General Procedure for the Preparation of 3,3-difluoro-1-propylindoline-2-one (**14a–e**)

Briefly, 2.9 mmol (1 eq) of the corresponding isatins, **13a–e**, 4.7 mmol (1.63 eq) of propargyl bromide, 5.5 mmol (1.9 eq) of potassium carbonate, 0.9 mmol (0.32 eq) of sodium iodide and 6 mL of anhydrous DMF were added in a monotube flask. The progress of the reaction was monitored using TLC (hexane/ethyl acetate 7:3). After 24 h at room temperature, 5 mL of 3N HCl solution and ice were added, and a precipitate was formed. This solid was filtered with a vacuum to give the desired product.

The attributes can be summarized as follows:

**3,3-difluoro-1-propylindoline-2-one (14a)**: Yield: 85%; m.p.: 76–78 °C (lit. 76–78 °C) [19]. IR (cm^−1^): 3250; 2125; 1744; 1304. MS-ESI [M+Na]^+^ (%): 230 (100). MS-GC (70 eV, *m*/*z*, %): 168 (100); 207 (78); 179 (18); 178 (12); 126 (12). 

**3,3-difluoro-5-methyl-1-propylindoline-2-one (14b)**: Yield: 94%; m.p.: 102–105 °C (lit. 102–104 °C) [19]. IR (cm^−1^): 3247; 1748; 1298. MS-GC (70 eV, *m*/*z*, %): 221 (94); 183 (100); 192 (17); 222 (13); 178 (13). 

**5-chloro-3,3-difluoro-1-propylindoline-2-one (14c)**: Yield: 77%; m.p.: 87–89 °C (lit. 87–89 °C) [17]. IR (cm^−1^): 3259; 2126; 1726; 1335. MS-GC (70 eV, *m*/*z*, %): 202 (100); 241 (92); 243 (32); 178 (26).

**5-bromo-3,3-difluoro-1-propylindoline-2-one****(14d)**: Yield: 52%; m.p.: 115–116 °C. IR (cm^−1^): 3261; 1726; 1276. MS-ESI [M+1]^+^ (%): 284 (100). NMR^1^H (400 MHz; CD_3_COCD_3_, δ, ppm): 3.42 (t, *J* = 8.8 Hz, 1H, H-3′); 4.60 (d, *J* = 2.5 Hz, 2H, H-1′); 7.30 (d, *J* = 8.5 Hz, 1H, H-7); 7.90 (dt, *J* = 1.1 Hz, H-6); 8.0 (d*, J =* 1.8 Hz, 1H, H-4). NMR^13^C (100 MHz; CD_3_COCD_3_, δ, ppm): 29.6 (C-1′); 75.6 (C-3′); 76.5 (C-2′); 110.2 (t, *J* = 248.6Hz, C-3); 113.4 (C-7); 116.0 (C-5); 120.6 (t, *J* = 22.9 Hz, C-3a); 127.8 (C-4); 137.0 (C-6); 141.4 (t, *J* = 6.8 Hz, C-7a); 163.1 (t, *J* = 29.9 Hz, C-2). NMR^19^F (376 MHz; DMSO, δ, ppm): −110.6.

**5-fluoro-3,3-difluoro-1-propylindoline-2-one****(14e)**: Yield: 62%. IR (cm^−1^): 3305; 1753; 1293; 1087. MS-ESI [M+1]^+^ (%): 226 (100). MS-GC (70 eV, *m*/*z*, %): 186 (100); 225 (78); 158 (16); 144 (15); 196 (12). NMR ^1^H (400 MHz; CD_3_COCD_3_, δ, ppm): 3.41 (m, 1H, H-3′); 4.61 (d, *J* = 2.5 Hz, 2H, H-1′); 7.37 (m, 1H, H-7); 7.57 (m, 1H, H-6); 7.8 (d, *J* = 1.8 Hz, 1H, H-4). NMR ^13^C (100 MHz; CD_3_COCD_3_, δ, ppm): 29.5 (C-1′); 75.4 (C-3′); 76.5 (C-2′); 112.8 (t, *J* = 243.9 Hz, C-3); 113.1 (C-7); 119.8 (td, *J* = 8.7 Hz, C-4); 120.8 (d, *J* = 23.7 Hz, C-3a); 138.2 (td, *J* = 2.0 Hz, C-6); 157.6 (t, *J* = 2.5 Hz, C-5); 160.0 (t, *J* = 2.6 Hz, C-7a); 163.4 (t, *J* = 29.7 Hz, C-2). NMR ^19^F (376 MHz; DMSO, δ, ppm): −110.6.

### 3.2. Biological Evaluation

#### 3.2.1. Cell Lines

The A549 cell line (CCL-185™; ATCC, Manassas, VA, USA), from human lung carcinoma epithelium, and the K562 cell line (Rio de Janeiro Cell Bank, BCRJ: 0126, Rio de Janeiro, RJ, Brazil), a human chronic myeloid cell line expressing the BCR-ABL1 protein, were grown in RPMI-1640 culture medium (Merck, Darmstadt, Hesse, Germany) supplemented with 10% heat-inactivated fetal bovine serum (FBS; Vitrocell, Campinas, SP, Brazil). WSS-1 [WS-1] (ATCC CRL-2029™) human kidney epithelial cells were grown in high-glucose DMEM (Vitrocell) supplemented with 10% FBS (Vitrocell). All cell lines were maintained at 37 °C under 5% CO_2_, in a water jacket CO_2_ incubator (Forma Series II incubator, Thermo Fisher Scientific, Waltham, MA, USA).

#### 3.2.2. Cytotoxicity Assays

The cytotoxic activity of imatinib and its analogs was evaluated by a fluorimetric assay, based on the metabolic reduction of resazurin to resorufin. Briefly, cells were seeded in 96-well microplates (2 × 10^4^ cells/well for K562 cells and 5 × 10^4^ cells/well for A549 and WSS-1 cells) and incubated with 10 µM of the test compounds for 48 h at 37 °C. A resazurin (Merck, Darmstadt, Hesse, Germany) solution (final concentration: 0.01 mg/mL) was added to each well before the end of the experiment, at 1 h before concluding for K562 cells, and 2 h before concluding for A549 and WSS-1 cells. Fluorescence readings (λ_ex_ = 560 nm, λ_ex_ = 590 nm) were carried out immediately after the addition of resazurin (t_0_) and at the end of the experiment (t_n_) with a FlexStation 3 Benchtop multimode microplate reader (Molecular Devices, San Jose, CA, USA). Resorufin fluorescence was calculated by subtracting the fluorescence measured at t_0_ from that at t_n_. Cell viability was expressed as percentages, considering the mean resorufin fluorescence of the control wells (cells incubated with only DMSO) to be 100%. Tests were performed in triplicate. The same assay was carried out to generate 8–10 points concentration-response curves (300 µM top compound concentration). From this analysis it was possible to calculate the IC_50_ (concentration of a compound the reduces in 50% cell viability) values.

#### 3.2.3. Kinase Inhibition Assays

The ABL1 inhibitory activity assay was performed using the ABL1 Kinase Enzyme System and ADP-Glo™ Kinase assay kit (Promega, Madison, WI, USA) according to the manufacturer’s instructions. The general procedure was as follows: kinase (2.5 ng/reaction) was incubated with the substrate (1 μg), compound (0.5 and 10 μM), and ATP (25 μM) in a commercial buffer solution with a reaction volume of 5 μL. In every experiment, no-enzyme and no-compound control reactions were included to represent the background luminescence (0% activity) and uninhibited kinase activity (100% activity), respectively. The assay 384-well plate was incubated at room temperature for 1 h. Afterward, 5 μL of ADP-Glo reagent was added to each well to stop the reaction and consume the remaining ADP within 40 min. At the end of this period, 10 μL of kinase detection reagent was added to each well and incubated for 30 min to produce a luminescence signal. Kinase activity assays were performed in triplicate at each inhibitor concentration. The luminescence was measured in a FlexStation 3 (Molecular Devices, San Jose, CA, USA) plate reader. The percentage of kinase activity (KA) was calculated by subtracting the average no-enzyme control luminescence values from all kinase-containing reactions with or without the compound, then converting these net luminescence values into percentage activity, based on the no-compound control reactions representing 100% kinase activity. Inhibition (I, %) was calculated as 100%-KA. Imatinib at 10 μM was used as a positive inhibition control.

#### 3.2.4. Statistical Analysis

IC_50_ values were calculated by a four-parameter logistic curve function, using GraphPad Prism version 6.01 for Windows (GraphPad Software, San Diego, CA, USA, www.graphpad.com) (accessed on 29 December 2021).

## 4. Conclusions

Many patients have presented with resistance to TKIs, which has led to the search for new therapeutic options. In this work, ten new imatinib analogs were synthesized, and their antiproliferative activities against cancer (K562 and A549) and normal (WSS-1) human cell lines were evaluated. A molecular hybridization strategy was used to obtain compounds **2a****–****e** and **3a****–****e** with PAPP and isatin scaffold units, spaced with a 1*H*-1,2,3-triazole ring and with different substituents at the C-5 carbon of isatin. Biological evaluations showed that compound **3b** exhibited the highest cytotoxic activity against the K562 cell line, with an IC_50_ value of 35.8 µM. Evaluations of the A549 cell line showed that **3a**, **3c** and **3d** were more potent and selective than imatinib. These results indicate the importance of the CF_2_ group since, in compounds **2a****–****e**, the absence of this group led to a loss of activity. These data suggest that these compounds can be used as prototypes for other studies of compounds against CML and lung cancer. The imatinib analogs (**2a****–****e** and **3a****–****e**) were also investigated for the inhibition of ABL1 enzyme activity and did not show inhibitory effects at all concentrations. In the future, other enzymatic studies will be carried out, including work with the BCR-ABL1 enzyme.

## Data Availability

Not applicable.

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
