# Peer review of "New Imatinib Derivatives with Antiproliferative Activity against A549 and K562 Cancer Cells"

_molecules, 2022, doi:10.3390/molecules27030750_

Round 1
Reviewer 1 Report
Title:
The title covers the central subject of the paper and no changes are necessary.
Abstract:
The introductory part of the abstract no has explained the problem well enough to answer the questions raised. Shortly write a justification for the research.
The methods of the work are not well outlined and does not explain how the work was conducted.
The results can be better explained in more detail. Giving importance also to the synthesis of compounds.
Introduction
Between lines 47 to 52 there is a large paragraph where there could be more citations or there could be the distribution of citations, specifically when it says, “many cases of resistance…” Cite others research.
Overall, needs to be rewritten described the hypothesis/objective of the study, to make clear the objective of the work.
Methodology
The material and methods should precede the results and discussion section. Check the submission guidelines.
In the section 3.2.2 Cytotoxicity tests: introduce the treatment concentrations about the compounds utilized.
In the titles and subtitles of lines 234 and 235 there should be section numbering. Revise this and the other titles and subtitles.
Results and Discussion
In general, it does not need be enhanced. Showed the results in the objective way.
Author Response
Point 1: Abstract: The introductory part of the abstract no has explained the problem well enough to answer the questions raised. Shortly write a justification for the research.
Response: We appreciate the comments and the careful revision of our manuscript. We want to mention that changes were made in order to make this aspect more straightforward for the reader. Specifically, we added the following paragraph: “Tyrosine kinase enzymes are among molecular targets for the treatment of some human neoplasms such as lung and chronic myeloid leukemia. Mutations in the enzyme domain can cause resistance, and new inhibitors capable of circumventing these mutations are highly desired”
Point 2: Abstract: the methods of the work are not well outlined and does not explain how the work was conducted.
Response: We want to mention that changes were made in order to make this aspect more straightforward for the reader. We rewrote the assay part as follows ”All of them were tested against A549 and K562 cells, lung cancer and chronic myeloid leukemia (CML) cell lines, respectively. In A549 cells, the 3,3-difluorinated compound (3a), the 5-chloro-3,3-difluorinated compound (3c) and the 5-bromo-3,3-difluorinated compound (3d) showed IC50 values of 7.2, 6.4, and 7.3 μM, respectively and were all more potent than imatinib (IC50 of 65.4 μM). In K562 cells, the 3,3-difluoro-5-methylated compound (3b) decreased cell viability to 57.5% at 10 µM showed IC50 value of 35.8 μM (imatinib, IC50 = 0.08 μM). The results suggest that 3a, 3c, and 3d can be used as prototypes for the development of more potent and selective derivatives against lung cancer.”
Point 3: Abstract: the results can be better explained in more detail. Giving importance also to the synthesis of compounds.
Response: We agree with the reviewer's comments and Due to the reduced space for the abstract, we decided to insert only one phrase: “The 1,2,3-triazole core was used as a spacer in the derivatives through click chemistry reaction in good yields”.
Point 4: Introduction: Between lines 47 to 52 there is a large paragraph where there could be more citations or there could be the distribution of citations, specifically when it says, “many cases of resistance…” Cite others research
Response: Done. We agree with the reviewer's comments and we have now added more references to this aspect.
Point 5: Overall, needs to be rewritten described the hypothesis/objective of the study, to make clear the objective of the work.
Response: We appreciate the comments and the careful revision of our manuscript. We want to mention that the introduction was rewritten and changes were made in order to make this aspect more straightforward for the reader. Specifically, we added the following paragraph: “The requirement imposed by frequent mutations occurring in the kinase domain is still among the greatest limitations in treatment by target therapy, especially in oncology [11]. Therefore, it is imperative to develop new tyrosine kinase inhibitors (TKIs) that can reduce known resistances and others that may arise.”
Point 6: Methodology: The material and methods should precede the results and discussion section. Check the submission guidelines.
Response: We appreciate the reviewer's observation, but the submission guidelines at https://www.mdpi.com/journal/molecules/instructions#preparation the sequence: Introduction, Results, Discussion, Materials and Methods, Conclusions and Patents. Furthermore, we took care to compare this work with other similar publications and, the sequence was in agreement.
Point 7: In the section 3.2.2 Cytotoxicity tests: introduce the treatment concentrations about the compounds utilized.
Response: This information was included in the text as suggested by the reviewer.
Point 8: In the titles and subtitles of lines 234 and 235 there should be section numbering. Revise this and the other titles and subtitles.
Response: Done.
Reviewer 2 Report
The paper presents the synthesis and antiprpliferative activity of two series of imatinib analogs. These compounds were designed as molecular hybrids containing phenylamino-pyrimidine pyridine and isatin scaffolds connected by a 1H-1,2,3-triazole ring. The cytotoxic activity of compounds was evaluated in two human cancer cell lines (K562 64 and A549) as well as in normal cell line (WSS-1). Only some 3,3-difluoro-1-((1-(4-methyl-3-((4-(pyridin-3-yl)pyrimidin-2-yl)amino)phenyl)-1H-1,2,3-triazole-4-yl)methyl)indoline-2-ones - containing the CF2 group - showed cytotoxicity and selectivity. None of the analogs inhibited the activity of the ABL1 enzyme.
Comments:
Lines 147-150: text should be moved to section 2.2.
Figures 1 and 2: it is difficult to accurately read the percentage of cell viability (scale too small). This should be corrected or the results should be presented in a table.
Paragraphs 2.2.1. and 2.2.2.: As shown in Figures 1 and 2, the cytotoxicity of the compounds was assessed in the cell viability assay, so the test results should be related to cell viability and not to inhibition of cell viability calculated as 100% -% of cell viability.
Lines 168 and 194: strains should be replaced with cell lines or cells.
Lines 204-208: this information is included in the paragraph 3.2.3.
The authors did not include the 1H and 13C NMR spectra in the Supplementary material.
Author Response
Comments (Reviewer 2)
Point 1: Lines 147-150: text should be moved to section 2.2.
Response: Done.
Point 2: Figures 1 and 2: it is difficult to accurately read the percentage of cell viability (scale too small). This should be corrected or the results should be presented in a table.
Response: Done.
Point 3: Paragraphs 2.2.1. and 2.2.2.: As shown in Figures 1 and 2, the cytotoxicity of the compounds was assessed in the cell viability assay, so the test results should be related to cell viability and not to inhibition of cell viability calculated as 100% -% of cell viability.
Response: The results expressed as “% viability inhibition” were substituted by “% viability” in the following parts of the paper: Abstract, Introduction, Figure 1 and Results and discussion section.
Point 4: Lines 168 and 194: strains should be replaced with cell lines or cells.
Response: Done.
Point 5: Lines 204-208: this information is included in the paragraph 3.2.3.
Response: The passage that describes the protocol of the kinase inhibition assay was removed from Results and discussion section (2.3 Kinase inhibition assay) as suggested by the reviewer. The paragraph was edited.
Point 6: The authors did not include the 1H and 13C NMR spectra in the Supplementary material.
Response: Done.